# Targeting HGF/c-MET Axis in Pancreatic Cancer

**DOI:** 10.3390/ijms21239170

**Published:** 2020-12-01

**Authors:** Srinivasa P. Pothula, Zhihong Xu, David Goldstein, Romano C. Pirola, Jeremy S. Wilson, Minoti V. Apte

**Affiliations:** 1Pancreatic Research Group, South Western Sydney Clinical School, Faculty of Medicine and Ingham Institute for Applied Medical Research, Liverpool, NSW 2170, Australia; s.pothula@unsw.edu.au (S.P.P.); Zhihong.xu@unsw.edu.au (Z.X.); r.pirola@unsw.edu.au (R.C.P.); js.wilson@unsw.edu.au (J.S.W.); 2Faculty of Medicine, The University of New South Wales, Sydney, NSW 2052, Australia; d.goldstein@unsw.edu.au

**Keywords:** HGF-c-MET, Pancreatic Cancer, Stromal-tumour interactions

## Abstract

Pancreatic cancer (pancreatic ductal adenocarcinoma (PDAC/PC)) has been an aggressive disease that is associated with early metastases. It is characterized by dense and collagenous desmoplasia/stroma, predominantly produced by pancreatic stellate cells (PSCs). PSCs interact with cancer cells as well as other stromal cells, facilitating disease progression. A candidate growth factor pathway that may mediate this interaction is the hepatocyte growth factor (HGF)/c-MET pathway. HGF is produced by PSCs and its receptor c-MET is expressed on pancreatic cancer cells and endothelial cells. The current review discusses the role of the MET/HGF axis in tumour progression and dissemination of pancreatic cancer. Therapeutic approaches that were developed targeting either the ligand (HGF) or the receptor (c-MET) have not been shown to translate well into clinical settings. We discuss a two-pronged approach of targeting both the components of this pathway to interrupt the stromal–tumour interactions, which may represent a potential therapeutic strategy to improve outcomes in PC.

## 1. Introduction

The five-year survival rate of pancreatic cancer (PC) in the United States is currently 9%, an improvement from the less than 5% survival rate ten years ago [1]. However, it is still the fourth leading cause of cancer-related death in both men and women [1] and is predicted to become the second leading cause by 2030 [2].

The clinical outcome of PC remains dismal largely due to the lack of early detection and limited treatment options upon diagnosis. Most patients (>80%) have metastatic disease at diagnosis and are therefore not suitable for surgical treatment. Only about 15–20% of patients are deemed to have resectable tumours at diagnosis, but even after surgical removal, a significant proportion of patients develop recurrence. Histologically, PC is characterised by a prominent desmoplastic/stromal reaction, which has only received attention in the last two decades (Figure 1). This stromal reaction is made of cellular components such as pancreatic stellate cells (PSCs), immune cells, endothelial cells, neural elements, and non-cellular components such as collagens, fibronectin, glycoproteins, proteoglycans, hyaluronic acid, cytokines, growth factors, and serine protein acidic, and rich in cysteine (SPARC) [3]. This collagenous stroma is produced by pancreatic stellate cells (PSCs), which are resident cells of the pancreas normally comprising 4–8% of all pancreatic parenchymal cells. 

PSCs become activated in diseased states (necroinflammation or cancer) and synthesise excessive amounts of extracellular matrix proteins, leading to the fibrosis of chronic pancreatitis as well as the desmoplasia of pancreatic cancer [5]. In vitro and in vivo studies have established the existence of a bidirectional interaction between PSCs and cancer cells which can promote cancer progression [6]. Pancreatic cancer cells have been shown to stimulate PSC proliferation, migration, and extracellular matrix (ECM) production. In turn, PSCs induce cancer cell proliferation and colony formation while at the same time decreasing apoptosis, thus increasing cancer cell survival. In addition, PSCs stimulate cancer cell migration/invasion, stemness and epithelial-to-mesenchymal transition (EMT), effects which facilitate metastasis and the recurrence of pancreatic cancer [5,7,8,9,10,11]. Interestingly, supporting this active role of PSCs in PC metastasis, an earlier study by our group using an orthotopic model of pancreatic cancer reported the world-first finding that PSCs from the primary tumour can travel to distant metastatic sites, where they likely facilitate the seeding and growth of cancer cells [11]. 

PSCs also interact with other stromal components of PC such as endothelial cells, immune cells, neuronal cells, and extra cellular matrix (ECM) components [6]. PSCs play both proangiogenic and antiangiogenic roles in PC (see review [12]). PSCs regulate endothelial cell proliferation and migration thus modulating angiogenesis [13]. They are known to secrete potent proangiogenic growth factors, such as vascular endothelial growth factor (VEGF) and fibroblast growth factor (FGFs such as FGF2, FGF5 [14]). PSCs express FGF2 and when co-cultured with cancer cells, this mRNA expression in enhanced (unpublished data). Conditioned media from these cells have been shown to stimulate tube formation by human microvascular endothelial cells (HMEC-1) and this effect was mainly mediated by VEGF [11]. PSCs have also been shown to produce vasohibin-1 and endostatin exerting antiangiogenic effects [15].

The microenvironment of PC is immunosuppressive (see review [12]). PSCs have been shown to interact with immune cells in the stroma in different ways. PSCs may enable sequestration of circulating cytotoxic CD8 +ve T cells in the stroma via secretion of the chemokine CXCL12 [16]. PDAC patient T cells express higher levels of CXCL12 receptor (CXCR4) than those of healthy donor T cells, demonstrating the role of CXCL12/CXCR4 axis in this function. This impedes the destruction of cancer cells by the cytotoxic cells, thus facilitating immune evasion in PC. Activated PSCs produce VEGF, platelet-derived growth factor (PDGF) and inflammatory mediators (cytokines and chemokines) and induce the recruitment of circulating macrophages [17,18]. Interestingly, IL4 and IL13 produced by activated PSCs have been shown to drive macrophages to the profibrogenic anti-inflammatory M2 phenotype [19], which contributes to the immunosuppressive milieu of pancreatic cancer.

With regard to neural invasion in pancreatic cancer, PSCs have been reported in stimulate the migration of PC cells along nerve axons, an effect mediated by the Sonic hedgehog (SHh) pathway. [20]. PSCs may also play a role in the pain of PC via the production of neurotrophic factors such as brain derived neurotrophic factor (BDNF) and nerve growth factor (NGF) [21].

Overall, a number of growth factors, (e.g., VEGF, PDGF, and FGF) mediate tumour–stromal interaction and PC progression. The receptors for these growth factors belong to a family of tyrosine kinase receptors. Several inhibitors of these receptors as well as growth factor neutralising antibodies/binding proteins have been developed, but with limited efficacy in clinical translation. The hepatocyte growth factor and its receptor, c-MET play an important role in cancer progression (as discussed in detail below), and this pathway warrants investigation for its role in pancreatic cancer.

## 2. Hepatocyte Growth Factor (HGF)/c-MET Pathway

As noted above, a major candidate factor regulating stromal-tumour interactions in pancreatic cancer is the hepatocyte growth factor (HGF) also known as scatter factor (SF). HGF is a 90 kDa glycoprotein produced by stromal cells of mesenchymal origin as an inactive single chain polypeptide (precursor/ pro-HGF). It is converted to its active hetero-dimeric form (consisting of α chain of 69 kDa and β chain of 34 kDa) by proteolytic cleavage. The α subunit contains an N-terminal hairpin loop and four kringle domains, and the β subunit includes a serine protease homology domain lacking enzymatic activity [22]. Active HGF is the sole ligand for an epithelial tyrosine-kinase receptor called c-MET. c-MET is a heterodimeric protein made of two disulphide-linked chains of 50 kDa and 145 kDa [23].

The binding of HGF to its receptor leads to dimerisation and phosphorylation of c-MET and subsequent activation of several signalling pathways including MAPK and PI3K that regulate proliferation, invasion, and migration of cancer cells. This pathway is particularly important in embryogenesis and organogenesis. This activation triggers a complex intracellular signal program supporting invasive growth [24] and is responsible for cell scattering, motogenesis, and survival [25]. MET activation in epithelial cells leads to acquisition of polarity as well as tubule formation [26,27]. HGF is reported to induce a re-organisation of microtubules, actin fibres and focal adhesion components (via intracellular GAB1, Akt, ERK signalling), promoting the proliferation and migration of epithelial cells towards healing wound edges [28].

Overall, MET activation (via further downstream signalling) is crucial during embryonic development [29,30,31] as well as in adult life due to its role in tissue homeostasis, wound healing, and organ regeneration [25].

## 3. HGF/c-MET Pathway in Pancreatic Cancer

Though HGF/c-MET signalling plays an important role in embryonic development and wound healing, this pathway is rarely active in adults apart from malignancies. Dysregulated HGF/c-MET activation can be caused by genetic mutations, gene amplifications, protein overexpression, or a ligand-dependent autocrine or paracrine signalling loop [32,33,34].

The critical role of HGF/c-MET pathway in exerting mitogenic and motogenic effects on tumour cells has been reported in several cancers. Upregulation of HGF and/or c-MET expression in lung, breast, bladder, cervix, head and neck, stomach and blood were found to be associated with poor prognosis [35,36,37]. In colon cancer, HGF secreted by fibroblasts has been shown to drive colon cancer cell proliferation through c-MET dependent signalling [38]. Similar observations were made in pancreatic cancer wherein HGF was found to be secreted predominantly by PSCs and the receptor c-MET was expressed by cancer cells and endothelial cells [8,39].

In pancreatic cancer patients, elevated serum HGF levels have been reported to correlate with disease progression [40,41], and tumour expression of c-MET to be associated with poor survival [42]. The phosphorylation of c-MET has been described in patients with early distant metastases even after complete surgical resection of local disease [43]. c-MET activation has been reported to increase: resistance to gemcitabine therapy [44]; tumour cell motility [45]; and secretion of angiogenic factors [46] in pancreatic cancer. These effects are likely mediated via the activation/phosphorylation of downstream signalling such as PI3K/Akt, MAPK/ERK, or FAK. [44,45,47,48,49,50].

Given that HGF is the only known ligand for c-MET [36] and that both HGF and c-MET are associated with aggressive disease [29,51], there has been an increased interest in targeting this pathway [8,52]. The significant role of the HGF/c-MET pathway involved in the intricate crosstalk between the tumour and stromal compartments of PC was elucidated using an orthotopic model of PC by Pothula et al. [8] in 2016. Patient derived PSCs were mixed with AsPC-1 cells (human PC cell line) and implanted into the pancreas of Balb/c immunodeficient mice to produce orthotopic tumours. One week after the impantation of cells, when tumours had formed in the pancreas, mice were treated with the HGF neutralising antibody, AMG102 (Rilotumumab). This HGF neutralising antibody abrogated the crosstalk between stroma and tumour, resulting in reduced tumour size. Interestingly, when HGF inhibition was combined with c-MET inhibition and chemotherapy (triple therapy) in mouse models of early and advanced PC, a significant inhibitory effect was observed on cancer metastasis. In the early cancer model where treatment was instituted one week after the implantation of cancer cells + PSCs in mouse pancreas, metastasis was virtually eliminated. Even more importantly, in advanced cancer, where treatment was instituted 4 weeks after implantation of cancer cells + PSCs, (with well-formed tumours and metastatic lesions), metastasis was eliminated [10,53] (Figure 2). These findings were further validated in another mouse model that mimics the clinical scenario, where orthotopic tumours were resected and triple therapy was then tested as an adjuvant therapy [54]. 

Basilico and colleagues have found that concomitant MET/HGF targeting by a dual antibody/decoy strategy proved to be significantly effective in blocking MET activation both in vivo and in vitro [55]. Using a SCID mouse model of PC produced by orthotopic transplantation of a human pancreatic carcinoma that is engineered to express human HGF, the authors demonstrated that concomitant treatment with antibody and decoy significantly reduced metastatic spread. The promising results described above re-emphasise the importance of targeting both arms of HGF/c-MET pathway.

### 3.1. Role of HGF/c-MET Pathway in Hypoxia, Angiogenesis, Metastasis

Pancreatic cancer has a highly hypoxic microenvironment, in which hypoxia-inducible factor-1 α (HIF-1α) is activated. HIF-dependent pathways subsequently activate MET in pancreatic tumour cells [56,57], promote stromal-tumour interaction and induce neo-angiogenesis. Upregulation of c-MET expression on endothelial cells is reported to be required in in the early stages of angiogenesis, as the cells proliferate, change shape, and invade into the ECM layer [58].

As noted earlier, in vitro studies indicate that PSCs can interact closely with endothelial cells and aid in neo-angiogenesis [39]. Conditioned medium from human PSCs was found to induce proliferation and tube formation of human microvascular endothelial cells (HEMC-1). These effects were abrogated in the presence of a VEGF neutralising antibody, suggesting that VEGF secreted by PSCs may mediate the observed pro-angiogenic effects. Interestingly, HGF has also been shown to exert pro-angiogenic effects and has been reported to act synergistically with VEGF in this process [59,60]. In this regard, Patel et al. have reported that inhibition of the HGF/c-MET pathway using the HGF neutralising antibody AMG102 and the c-MET inhibitor PHA-665752 significantly decreased the inductive effects of PSCs on HMEC-1 proliferation and tube formation induced [39]. These inhibitory effects were associated with downregulation of the downstream signalling molecules ERK1/2 and p38.

Metastasis is a multifaceted process of tumour progression, that involves migration of cancer cells accompanied by the microenvironment such as stroma. From an intact primary tumour, cancer cells acquire the ability to disrupt cell adhesion, to invade through the ECM, spread through the circulatory/lymphatic system and to finally extravasate, seed and proliferate at a location other than their site of origin [59]. The HGF/c-MET axis plays a key role in many stages of the metastatic process, from cellular dissociation in the primary tumour to reassociation within metastatic sites. HGF influences destabilisation of cell-cell junctions, supports cytoskeletal remodelling, controls integrin functions and stimulates matrix metalloproteinase (MMP)-mediated proteolysis of ECM [24,60].

With regard to the movement of cancer cells across the endothelial cell barrier during metastasis, HGF facilitates cancer cell–endothelial cell contact through phosphorylation of focal adhesion kinase (FAK) [61] and reduces occludin, the primary protein in endothelial tight junctions [62]. This disruption in tight junctions causes morphological changes. HGF also increases permeability between vascular endothelial cells and promotes movement of cancer cells across the endothelial cell barrier into the adjacent tissues. Additionally, this HGF-induced FAK activation also upregulates MMP-1, MMP-9, MMP-14 (all of these MMPs play an important role in cell invasion, thus increasing invasiveness of cancer) in gallbladder, prostate, and liver cancers [63,64,65]. Recent studies have also demonstrated that PSCs facilitate perineural invasion of pancreatic cancer and this process is mediated via the HGF/c-MET pathway [66].

Most studies examining the in vitro effects of HGF itself on pancreatic cancer growth and invasion have been limited to cancer cells alone, and have not included stromal cells or PSCs [45,67,68]. However, Qian and colleagues have demonstrated in vitro that patient tumour derived fibroblasts expressing HGF could initiate an apparent invasion-stimulating response in pancreatic cancer cells with high expression of c-MET but not in cells with low expression of c-MET [69]. Another in vitro study demonstrated the invasiveness of a pancreatic cancer cell line (PK8) that was enhanced by conditioned medium from fibroblast cell line (MRC5) under hypoxic conditions; importantly this effect was reduced by neutralizing HGF in the conditioned medium [70]. PK8 cells exposed to conditioned media collected from HGF-expressing MRC5 cells showed a significant increase in matrix metalloproteinases ((MMPs) MMP-2, MMP-7, MT1-MMP) and c-MET levels, as well as a concordant increase in c-MET phosphorylation, leading to enhanced migration and invasion (70).

### 3.2. HGF/c-MET and uPA Feed Forward Loop

As discussed earlier, HGF is secreted as an inactive pro-form, which is cleaved by proteases to exert its biological function. Urokinase plasminogen activator (uPA) is one such protease that plays an important role in modulating the HGF/c-MET pathway [71]. Once HGF binds to c-MET on cancer cells, uPA production by cancer cells is induced, leading to activation of more pro-HGF to active HGF [71]. Thus, a continuous positive feed forward loop is established increasing the overall facilitatory effect of HGF on cancer progression (Figure 3) [53]. Supporting this concept is a recent report by Buckley et al. [72] demonstrating that a uPA inhibitor, 6-substituted hexamethylene amiloride derivative, can significantly reduce the incidence of metastasis in an orthotopic pancreatic cancer model. 

### 3.3. HGF/c-MET and Microenvironment pH

A supportive tumour microenvironment coupled with other factors leads to enhanced metabolism in tumour cells, resulting in a loss of the normally rigid control of extracellular and intracellular pH [73]. Cells in health have a low rate of glycolysis followed by pyruvate oxidation, while cancer cells produce energy by a high rate of glycolysis followed by lactic acid fermentation in the cytosol [74]. Cellular pH plays a regulatory role in numerous processes such as cell cycle, cellular motility, intracellular homeostasis, and subsequent malignant transformation [73,75,76,77]. A systemic abnormality of cellular acid–base homeostasis plays a key role in the transduction of intracellular signals of a wide array of growth factors. Sodium-proton exchangers (also known as NHEs) and other transporters maintain cytoplasmic pH and expel protons outside the cells creating an acidic extracellular pH [52,78] that exacerbates the invasive properties of transformed tumour cells [77].

HGF has been reported to activate NHEs directly, leading to lower extracellular pH [79], which can induce the trafficking of cathepsin-rich lysosomes to the cell periphery [80]. The extracellular release of cathepsins from lysosomes, and other proteases (such as urokinase plasminogen activator -uPA) [81] facilitates enhanced ECM proteolysis, migration, and invasion. A study in prostate cancer has revealed that HGF induces lysosome tracking to the periphery of the cell by phosphorylating the MET receptor and activating kinase cascades such as PI3K and Rho A GTPases [82]. This study also showed that HGF treatment resulted in increased microtubule accumulation at the cell surface protrusions coinciding with the localisation of lysosomes, NHE activity, and cathepsin B secretion, eventually leading to enhanced invasion by prostate tumour cells [82].

### 3.4. HGF/c-MET and Treatment Resistance

The HGF/c-MET pathway has been demonstrated to play a role in chemoresistance (particularly to gemcitabine). Aberrant MET activation is known to impart gemcitabine resistance (See review [25]). c-MET reportedly is a stem cell marker in pancreatic cancer [83]. MET expression was seen in progenitor cells, showing pro-survival (anti-apoptotic) signals [60]. Gemcitabine treatment possibly exerts its cytotoxic effects on susceptible cancer cells, leaving behind a resistant population of cancer cells with stem cell-like characteristics [8]. In support of this concept, several studies have demonstrated that gemcitabine resistant pancreatic cancer cells also express high levels of stem cell markers as well as EMT markers [84]. Notably, HGF/c-MET inhibition has been shown in preclinical models to increase drug delivery and improve chemotherapeutic response both in genetically engineered mouse models [85] and orthotopic models of pancreatic cancer [53,86]. More recently, Firuzi et al. have also shown that inhibiting HGF-c-MET pathway aided in overcoming drug resistance using their spheroid models [87].

HGF itself has recently been shown to cause MET inhibitor resistance in a paracrine manner [88]. MET inhibitors reduce proliferation, invasion, migration, and downstream signalling in gastric MET-amplified cancer cells, but overexpression of HGF in cancer cells impairs this phenomenon. It has also been shown that lactate (produced by MET/EGFR TKI-resistant cancer cells) enhances the production/secretion of HGF by cancer-associated fibroblasts, which in turn activates MET-dependent signalling pathways in cancer cells, causing adaptive resistance to MET inhibitors [89]. Cancer-associated fibroblasts also reprogram the ECM to retain more paracrine HGF, which in turn positively affects the activation of c-MET and its downstream signaling pathways, thereby causing further resistance to MET inhibitors [89].

MET activation has been further demonstrated to influence macrophage polarisation, shifting its phenotype from an immunological active M1 phenotype to a trophic, growth-stimulating M2 state, thus promoting an immunosuppressive microenvironment [60,90].

### 3.5. Role of HGF as a Diagnostic Marker

Lee and colleagues [91] studied inflammatory mediator proteins (IMPs) using a multiplexed IMP-targeted microarray in pancreatic cyst fluid obtained during endoscopic ultrasound fine needle aspiration (EUS-FNA). They compared IMP profiles in pancreatic cyst fluid from branch duct intraductal papillary mucinous neoplasms (BD-IPMNs) and inflammatory cysts. They have been able to demonstrate that HGF was highly expressed in inflammatory cystic fluid in contrast to the cystic fluid from IPMNs [91]. Hence, HGF expression could be used as a differential diagnostic marker.

## 4. Targeting HGF/c-MET Pathway

Given the weight of evidence (discussed above) in support of a critical role for the HGF/c-MET pathway in pancreatic cancer progression, it is reasonable to consider this pathway as a potentially useful therapeutic target for this disease.

There have been some encouraging data pertinent to HGF and/or c-MET inhibitors in clinical trials (mostly phase I/II trials) in several non-pancreatic cancers [92,93]. These reports have prompted targeting of the HGF/c-MET axis in PC treatment. However, information regarding such an approach in PC is scarce, with only two trials involving less than 10 patients being reported [94,95]. Many ongoing clinical trials in a variety of cancers, in phases I, II, and III have employed MET kinase inhibitors or monoclonal antibodies for MET (MAbs such as Onartuzumab from Genentech). Most advanced in clinical development among the c-MET targeted therapies is Tivantinib (ARQ197), a non-adenosine triphosphate (ATP)-competitive c-MET inhibitor, which is in phase III development for various malignancies [96]. Currently, it is also being used in a randomised phase II study (ARQ 197 vs. gemcitabine) in treatment-naïve PC patients with unresectable locally advanced or metastatic PDAC [97] Results from this study are not yet available. However, several recent phase III trials using these agents failed to inhibit HGF/MET signalling in gastric cancers. Such failures prompted detailed analyses of these studies leading to several explanations [98,99]. However, consensus across most of these was that two main factors were involved in the failure of HGF/MET-targeted drugs in clinical practice: the inappropriate selection of specific patient populations and the development of resistance to the MET-targeted drugs [100,101].

(i) Approaches targeting HGF

The ligand HGF is also an obvious therapeutic target in PC considering its significant role in promoting tumorigenesis in cases exhibiting MET mutation [102,103]. Michieli and colleagues showed that the transforming potential displayed by mutant forms of MET found in human cancer is sensitive to, and can be entirely dependent on, the availability of the ligand HGF [103]. Moreover, the mutant MET-induced transformation of cells in their study was inhibited by HGF antagonists and increased by HGF stimulation, supporting the concept of HGF as a potentially useful target molecule. 

Clinical trials with HGF monoclonal antibody (MAb) in combination with other chemotherapeutic drugs are currently underway [52]. HGF Mab therapies were investigated for various cancers and included Rilotumumab (AMG102) from Amgen [95,104], Ficlatuzumab from AVEO pharmaceuticals and HuL2G7 from Millennium pharmaceuticals (www.vai.org/metclinicaltrials). Another novel compound, NK4, an intra-molecular fragment of HGF, has been shown to have promising results in vitro and in vivo [105,106] by targeting the HGF/c-MET axis. However, the orthotopic models used in these studies did not have the stromal compartment, and hence did not adequately recapitulate human PC.

AMG102 or Rilotumumab is a fully humanised IgG2 monoclonal antibody (mAb) against human HGF that blocks the binding of HGF to its receptor and inhibits HGF/MET-mediated responses, including cell proliferation, survival and invasion [104,107]. However after Phase II/III trials in gastric cancer, this compound has been withdrawn from clinical development [98]. One of the postulated reasons for the failure of AMG102 is the inappropriate selection of the patient cohort. Another postulated reason is the less-than-optimal efficacy of the antibody possibly due to the fact that it is only a partial antagonist of HGF (binding the beta chain of HGF), but leaving the high affinity alpha chain free to bind to c-MET [108].

Recently, another HGF neutralising antibody, YYB101, has been developed with promising pre-clinical results that have led to clinical trials in patients with refractory solid tumours [109]. Interestingly, YYB102 binds to the alpha chain of HGF thus more efficiently blocking HGF from binding c-MET, thereby leading to almost total inhibition of the signalling events downstream of the HGF-c-MET complex. In addition, YYB101 has 10–100-fold higher affinity for HGF than AMG102, making it a potentially superior HGF neutralising antibody when compared to AMG102 [109]. However, YYB101 has not yet been clinically tested in PC. It is currently in Phase1b/2a clinical trial in metastatic colorectal cancer (mCRC) patients in Korea.

(ii) Approaches using c-MET inhibitors

MET-specific small-molecule tyrosine kinase inhibitors (TKIs) are divided into two functionally distinct classes: type I (e.g., crizotinib) and type II (e.g., cabozantinib) inhibitors, which preferentially bind to the active and inactive conformations of MET, respectively.

(iii) Cabozantinib (XL184)

Cabozantinib is a potent, orally bioavailable, multitargeted small-molecule inhibitor of c-MET and VEGFR-2. The clinical efficacy of cabozantinib (trade name Cometriq, also known as XL184) in multiple tumour types is associated with extensive induction of cancer cell apoptosis as well as disruption of tumour vasculature and invasiveness, thereby blocking metastasis.

The therapeutic potential of this agent was studied in vitro by Hage et al. [44] in 2013. There was increased efficacy of gemcitabine even in high-gemcitabine-resistant PC cells and in patient-derived primary spheroidal cultures enriched in cancer stem cell markers [44]. Also, cabozantinib was found to inhibit SOX2, c-MET, and CD133 expression and the self-renewal potential of cancer cells. This compound, which is a potent dual inhibitor of c-MET and VEGFR-2 signalling, has been previously used in a transgenic model of pancreatic islet tumours [110] and an orthotopic model of PC in NOD SCID mice [83]. Both these studies reported favourable results such as inhibited tumour growth, reduced vasculature and tumour aggressiveness, and reduction of cancer-stem cell population as well as metastasis. Suppression of tumour invasion and metastasis by concurrent inhibition of c-MET and VEGF signalling (using cabozantinib) in pancreatic neuroendocrine tumours was also demonstrated by Sennino and colleagues [111].

The clinical efficacy of cabozantinib in several tumor entities is under investigation in randomised phase II studies, including patients with metastatic pancreatic cancer [112]. A randomised, double-blinded Phase III study of cabozantinib vs placebo in patients with advanced neuroendocrine tumours after progression on prior therapy (CABINET) is recruiting and is currently underway [113].

(iv) Crizotinib

This ATP-competitive c-MET inhibitor has been used by Avan and his colleagues [47] to demonstrate decreased tumor volume, prolonged survival, and increased blood and tissue concentrations of gemcitabine in orthotopic models of PC. They used double bioluminescent patient-derived orthotopic mouse PDAC models, which could be imaged longitudinally [47]. A synergistic interaction of crizotinib with gemcitabine has been reported on growth of primary PDAC cells in vitro, and primary tumour growth in vivo, but the effects on metastatic spread are unclear [47]. There were several later studies which prompted clinical trials including studies related to PC. In PC, crizotinib was specifically shown to inhibit peritoneal dissemination via suppressing HGF/MET signaling and RhoA activation [114].

During its clinical development, it was found that while crizotinib showed initial efficacy, patients inevitably acquired resistance to the drug. Several clinical reports described secondary MET mutations as mechanisms for crizotinib resistance [115,116,117,118], see Review [119] One such report by Zhang and colleagues also discussed the case of a patient who simultaneously acquired four rare resistance mutations (G1163R, D1228H, D1228A, and Y1230H) after the development of crizotinib resistance [115].

(v) Capmatinib (INC280)

Capmatinib is a potent and highly selective MET inhibitor, including the mutant variant produced by exon 14 skipping as seen with crizotinib. This c-MET inhibitor has been examined in an in vivo model of PC as well as in vitro [86]. Brandes and colleagues [86] used a xenograft orthotopic mouse model and syngeneic orthotopic models of PC and demonstrated reduced motility of PC cells with a 30% lymph node involvement in the treatment group when compared to 60% involvement in the control group, suggesting potential suppression of metastasis. Additionally, they looked at various PC cell lines (human and murine) to confirm the expression of c-MET as well as responsiveness to the inhibitor (INC280). They have reported that c-MET inhibition reduced HGF-induced PC cell proliferation as well as migration, at least in part, via inhibition of Akt, ERK and FAK phosphorylation. HGF-induced endothelial cell motility was strongly reduced by INC280. The authors reported impaired tumor growth, and improved efficacy when used in combination with gemcitabine [86]. However, their orthotopic model was produced by injecting PC cells alone and thus did not allow the study of the characteristic role of stroma regarding HGF/c-MET pathway. Metastatic spread was limited to lymph nodes and liver in this study; this might probably be due to the shorter time frame of treatment as well as lack of stromal influence.

Recently, Novartis announced that capmatinib (INC280), the first potential treatment for METex14 mutated advanced non-small cell lung cancer, had been granted priority FDA review. There are other phase II studies assessing the efficacy and safety of capmatinib monotherapy/combination therapy in patients with papillary renal cell carcinoma (NCT02019693), melanoma (NCT02159066) and solid tumours (NCT03040973).

Apart from these classes of MET inhibitors, anti-MET antibodies (emibetuzumab (LY2875358) and onartuzumab (MetMab)) have also been successfully applied in preclinical models of pancreatic cancer [49,120]. These antibodies are currently in clinical trials [121].

## 5. Conclusions

PC treatment remains a major unsolved problem, with existing therapies having limited success in terms of improving patient outcomes. Stromal-tumour interactions in pancreatic cancer are now widely recognised as major drivers of cancer progression. The HGF/c-MET pathway that mediates cross-talk between pancreatic stellate cells (the main collagen producing stromal cells) and cancer cells, as well as between PSCs and endothelial cells, is a potentially useful target for therapy. As detailed in this review, it is becoming increasingly clear that targeting just one or the other arm (ligand or receptor) of the HGF/c-MET pathway is an inadequate approach in PC. Instead, there is strong pre-clinical evidence to indicate that concurrent targeting of both the ligand and the receptor, combined with chemotherapy, offers the most effective approach for significantly reducing cancer progression in early as well as advanced settings of pancreatic cancer. Furthermore, HGF/c-MET inhibition itself appears to significantly inhibit recurrence in the adjuvant setting. Thus, we submit that a therapeutic strategy involving HGF/c-MET inhibition with or without chemotherapy, is eminently ready to be taken to clinical trials for pancreatic cancer in both neoadjuvant and adjuvant scenarios.

## Figures and Tables

**Figure 1 ijms-21-09170-f001:**
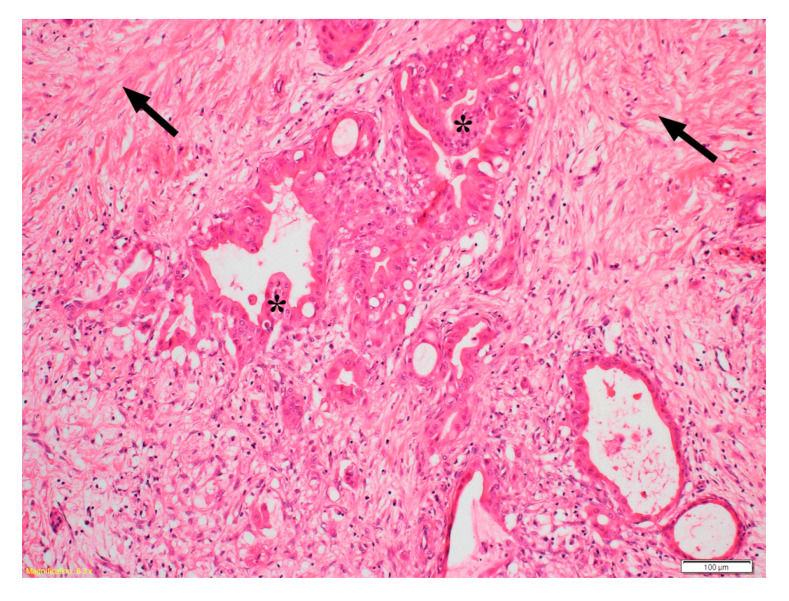
Representative photomicrograph of haematoxylin & eosin stained human pancreatic cancer section showing malignant elements (duct-like and tubular structures-indicated by asterisks) embedded in a highly fibrotic stromal reaction (indicated by arrows). Reprinted with permission from Elsevier [4].

**Figure 2 ijms-21-09170-f002:**
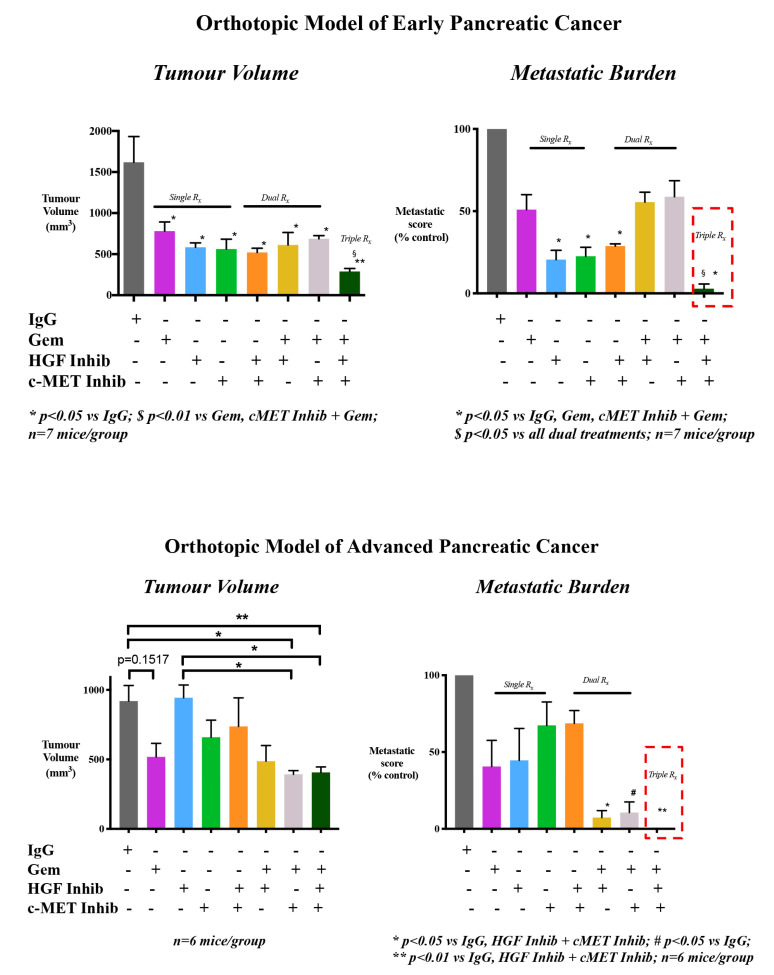
Effects of HGF/c-MET inhibition with and without gemcitabine on pancreatic cancer progression in orthotopic models of early and advanced pancreatic cancer. Pancreatic tumours were produced by injection of a mixture of human pancreatic cancer cells + human pancreatic stellate cells into the pancreas of Balb/c nude mice. For the early pancreatic cancer model, treatment was commenced one week after cell implantation, while for the advanced PC model treatment was commenced 4 weeks after cell implantation. In early pancreatic cancer, while single and dual combinations of the HGF neutralising antibody (HGF Inhib), c-MET inhibitor (cMET Inhib) and gemcitabine (Gem) reduced tumour volume compared to control (untreated) mice (IgG), the greatest reduction in tumour volume was observed in mice treated with the triple therapy. In advanced PC, the effects of single or dual agents on tumour volumes were variable, however, the greatest reduction in tumour volume was again observed in mice treated with triple therapy. Importantly, triple therapy had a striking effect on metastasis, with virtual elimination in early pancreatic cancer (only one liver nodule observed in one mouse) and complete absence of metastasis in advanced pancreatic cancer [10,53]).

**Figure 3 ijms-21-09170-f003:**
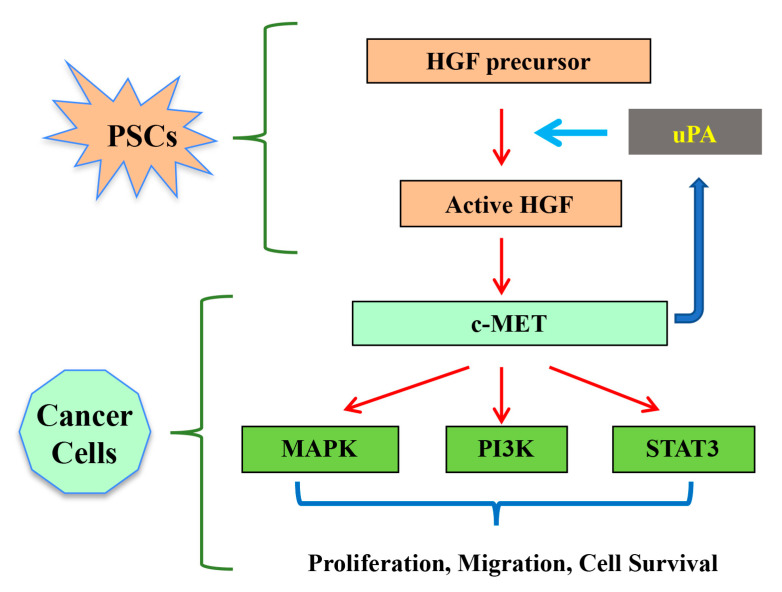
Schematic depiction of the possible role of HGF/c-MET pathway in pancreatic cancer. HGF secreted by PSCs in a precursor form is activated by proteolytic cleavage, mediated by proteases such as urokinase plasminogen activator (uPA). Upon binding of HGF to its receptor c-MET, downstream intracellular signalling cascades are activated which regulate cancer cell functions that influence tumour progression. This binding also increases production of uPA by cancer cells that further activates HGF from PSCs, thus forming a feed-forward loop. Adapted from Pothula et al. [53].

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
