# Peer review of "Targeting HGF/c-MET Axis in Pancreatic Cancer"

_ijms, 2020, doi:10.3390/ijms21239170_

Round 1

Reviewer 1 Report

The review by Pothula et al. well describes and comments on the relevant role of HGF/MET in pancreatic cancers; the main point of the cross talk between tumor and stroma, mediated by this ligand/receptor pair, is extensively discussed.

As a general comment, the review should not be contain an overview on all HGF/c-MET roles, but  it would be better if the authors emphasize the relevance of these activities specifically in relation to particular features of Pancreatic Cancer (PC). For example the paragraph on HGF/MET during organogenesis (lanes 117-135) can be skipped and the paragraphs HGF/c-MET and microenvironment pH (lanes 251-272) as well as the second half of the paragraph on resistance (lanes 285-296) should be more contextualized within the pancreas. Under this point of view, it is not relevant to include information on HGF/MET inhibitors in trials not focused on pancreatic cancer (in my opinion table 1 should be eliminated) and the part on HGF inhibitors should be reconsidered. On the other hand additional studies in preclinical setting on PC must be included and commented (one example is the use of MetMAb in the paper Jin et al., 2008 Cancer Res).

Minor points:

line 54-55 rewrite the sentence  “ In turn,  PSCs in turn induce cancer cell proliferation and …..” reporting in turn 2 times

line 73-76 please expand to better explain the mechanism by which PSCs act blocking cytotoxic activity of CD8+ cells; in this form the concept is not clear. Correct commas/full stops.

Line 244: reference on uPA as proHGF convertase is lacking.

The paragraph Role of HGF/c-MET pathway in hypoxia, angiogenesis, metastasis, is not well organized. It seems to be a collection of topics not connected one with the other. I suggest to touch each topics following the order of the title.

Line 205: Reference Patel et al., is not included in the text.

Line 293: reference on CAF/HGF/Met inhibitor resistance is lacking

Please always use italic for in vitro and in vivo.

Author Response

Response to reviewer # 1

We thank the reviewer for their positive and constructive comments on the submitted review.

(The review by Pothula et al. well describes and comments on the relevant role of HGF/MET in pancreatic cancers; the main point of the cross talk between tumour and stroma, mediated by this ligand/receptor pair, is extensively discussed).

(As a general comment, the review should not be contain an overview on all HGF/c-MET roles, but it would be better if the authors emphasize the relevance of these activities specifically in relation to particular features of Pancreatic Cancer (PC)).

(For example the paragraph on HGF/MET during organogenesis (lanes 117-135) can be skipped and the paragraphs HGF/c-MET and microenvironment pH (lanes 251-272) as well as the second half of the paragraph on resistance (lanes 285-296) should be more contextualized within the pancreas.)

We thank the reviewer for this comment. While we have removed the sections as suggested by the reviewer (indicated by strikethrough of text in the revised manuscript), we would be happy to defer to the Editor’s discretion as to whether the inclusion of our original text duplicates information presented in other articles in the Special Issue.

(Under this point of view, it is not relevant to include information on HGF/MET inhibitors in trials not focused on pancreatic cancer (in my opinion table 1 should be eliminated) and the part on HGF inhibitors should be reconsidered.)

We have now removed the Table 1 to make the review more focussed on pancreatic cancer. Regarding the HGF inhibitors, we have included inhibitors that worked in PDAC and gastric cancers due to the limited information on these inhibitors in pancreatic cancer scenarios.

(On the other hand additional studies in preclinical setting on PC must be included and commented (one example is the use of MetMAb in the paper Jin et al., 2008 Cancer Res).)

The paper by Jin et al had been referred to in another context in our manuscript (lines 207). However, as suggested by Reviewer 1, the following text has now been added to include the suggested study as follows “Apart from these classes of MET inhibitors, anti-MET antibodies (emibetuzumab (LY2875358) and onartuzumab (MetMab)) have also been successfully applied in preclinical models of pancreatic cancer. These antibodies are currently in clinical trials.

Minor points:

(line 54-55 rewrite the sentence “ In turn, PSCs in turn induce cancer cell proliferation and …..” reporting in turn 2 times)

We apologize for the typographical error; it has now been corrected.

(line 73-76 please expand to better explain the mechanism by which PSCs act blocking cytotoxic activity of CD8+ cells; in this form the concept is not clear. Correct commas/full stops.)

Apologies for the unclear sentence earlier, it has now been fixed as “PSCs may enable sequestration of circulating cytotoxic CD8 +ve T cells in the stroma via secretion of the chemokine CXCL12. PDAC patient T cells express higher levels of CXCL12 receptor (CXCR4) than those of healthy donor T cells, demonstrating the role of CXCL12/CXCR4
axis in this function. This impedes the destruction of cancer cells by the cytotoxic cells, thus facilitating immune evasion in PC.”

(Line 244: reference on uPA as proHGF convertase is lacking.)

This has now been added.

(The paragraph Role of HGF/c-MET pathway in hypoxia, angiogenesis, metastasis, is not well organized. It seems to be a collection of topics not connected one with the other. I suggest touching each topics following the order of the title.)

We believe that the phrase at the beginning of one of the paragraphs in the ‘metastasis’ section (line 221) may have led to the Reviewer’s comment. We have now modified this and the sections are organised in the order expected – hypoxia, angiogenesis, and metastasis.

(Line 205: Reference Patel et al., is not included in the text.)

This reference has now been added.

(Line 293: reference on CAF/HGF/Met inhibitor resistance is lacking)

This reference has now been added.

(Please always use italic for in vitro and in vivo.)

This has been corrected throughout the manuscript.

Reviewer 2 Report

I have enjoyed reading this literature study on the interesting and timely topic of molecular mechanisms underlying the activity of therapies targeting the HGF/MET axis in pancreatic ductal adenocarcinoma (PDAC)
The research question/thesis is complex, but the report addresses several viable research gaps and make an original overview of the information in this field of research
The main paragraphs and the discussion are appropriately summarizing the most recent literature. The text clearly guides the reader to the main findings by effectively using arguments, as well as some figures where appropriate.

Additionally, the authors discussed a two-pronged approach of targeting both the components of this pathway to interrupt the stromal-tumour interactions, which may represent a potential therapeutic strategy to improve outcomes in PDAC.

I would however suggest to following revisions:

1) Please explain better or delete the statement "Although a synergistic interaction of crizotinib with gemcitabine has been reported on growth of primary PDAC cells in vitro, the effect of treatments on primary tumour volumes and metastatic spread were not very well demonstrated" The cited article showed a reduction of tumor volume (Fig6B), so I am wondering why the authors reported that "primary tumor volumes and metastasic spread were not very well demonstrated"

2) please update the table 1 on: Therapies targeting HGF/c-MET pathway in Phase II/III clinical trials. Please add the most recent trias and the corresponding Clinicaltrials.gov numbers

3) please remove the underlined character with not-underlined in the paragraph: "Cabozantinib is a potent, orally bioavailable, multitargeted small-molecule inhibitor of c-MET and VEGFR-2. The clinical efficacy of cabozantinib (trade name Cometriq, also known as XL184) in multiple tumour types is associated with extensive induction of cancer cell apoptosis as well as disruption of tumour vasculature and invasiveness, thereby blocking metastasis."

4) please include among the preclinical studies reporting resistance to gemcitabine potentially mediated by the HGF/Met pathways the recent study published on Cancers on "Role of c-MET Inhibitors in Overcoming Drug Resistance in Spheroid Models of Primary Human Pancreatic Cancer and Stellate Cells"

5) Please improve the quality and readability of the figures 1 and 3. In particular in the figure 3 the dimension of the bar is not readable. Similarly in the figure 3 the text in the bar graphs is too small or faded.mortoever I do not understand why this figure is added after the references and why it was a legend before and another legend after the figure.

6) though this review is focused on PDAC, it also reports data on other tumor types and therefore i would strongly suggest the authors to include some more references about reviews the biology underlying the most recent success in the field of c-met pharmacology, i.e., the first MET tyrosine kinase inhibitors (TKIs) approved for clinical use (capmatinib) in NSCLC patients with METex14 mutations (Van Der Steen, et al. Journal Thoracic oncology 2016; Recondo et al., Cancer Discovery 2020).

Author Response

Response to Reviewer # 2:

1) Please explain better or delete the statement "Although a synergistic interaction of crizotinib with
gemcitabine has been reported on growth of primary PDAC cells in vitro, the effect of treatments on
primary tumour volumes and metastatic spread were not very well demonstrated" The cited article
showed a reduction of tumor volume (Fig6B), so I am wondering why the authors reported that
"primary tumor volumes and metastatic spread were not very well demonstrated"

We thank the Reviewer for pointing this out and have corrected the statement to “A synergistic
interaction of crizotinib with gemcitabine has been reported on growth of primary PDAC cells in vitro,
and primary tumour growth in vivo, but the effects on metastatic spread are unclear”.

2) Please update Table 1 on: Therapies targeting HGF/c-MET pathway in Phase II/III clinical trials.
Please add the most recent trias and the corresponding Clinicaltrials.gov numbers

In response to Reviewer #1’s comment to increase the focus of the review on pancreatic
cancer, we have now removed Table 1 which comprised clinical trials of HGF/c-MET
inhibition on non-pancreatic cancers.

3) Please remove the underlined character with not-underlined in the paragraph: "Cabozantinib is a
potent, orally bioavailable, multitargeted small-molecule inhibitor of c-MET and VEGFR-2. The clinical
efficacy of cabozantinib (trade name Cometriq, also known as XL184) in multiple tumour types is
associated with extensive induction of cancer cell apoptosis as well as disruption of tumour
vasculature and invasiveness, thereby blocking metastasis."

We apologise for this formatting error. This has now been corrected.

4)Please include among the preclinical studies reporting resistance to gemcitabine potentially
mediated by the HGF/Met pathways the recent study published on Cancers on "Role of c-MET
Inhibitors in Overcoming Drug Resistance in Spheroid Models of Primary Human Pancreatic Cancer
and Stellate Cells"

We thank the Reviewer for this suggestion. We have now modified the text as follows: “More
recently, Firuzi et al (97) have also shown that inhibiting HGF-c-MET pathway aided in
overcoming drug resistance in spheroid models of pancreatic cancer (97)”.

5) Please improve the quality and readability of the figures 1 and 3. In particular in the figure 3 the
dimension of the bar is not readable. Similarly in the figure 3 the text in the bar graphs is too small or
faded.mortoever I do not understand why this figure is added after the references and why it was a
legend before and another legend after the figure.

We apologise for the quality of figures. High quality (min 300dpi) figures had been originally
submitted to the journal. It appears that embedding of the Figures in a pdf version of the
manuscript for review purposes, significantly interfered with the quality of the Figures.,
High-quality images of Fig 1, Fig 2 and Fig 3 have been uploaded again for the Reviewer’s
interest.

6) Though this review is focused on PDAC, it also reports data on other tumor types and therefore i
would strongly suggest the authors to include some more references about reviews the biology
underlying the most recent success in the field of c-met pharmacology, i.e., the first MET tyrosine
kinase inhibitors (TKIs) approved for clinical use (capmatinib) in NSCLC patients with METex14
mutations (Van Der Steen, et al. Journal Thoracic oncology 2016; Recondo et al., Cancer Discovery
2020).

Given that Reviewer #1 suggested that we focus our review on pancreatic cancer, and as
noted in our response to point #2 above, we have now removed text pertaining to other
cancers.